# Low Disclosure Rates to Sexual Partners and Unsafe Sexual Practices of Youth Recently Diagnosed with HIV; Implications for HIV Prevention Interventions in South Africa

**DOI:** 10.3390/healthcare8030253

**Published:** 2020-08-03

**Authors:** Khensane Mengwai, Sphiwe Madiba, Perpetua Modjadji

**Affiliations:** Department of Public Health, School of Health Care Sciences, Sefako Makgatho Health Sciences University, Pretoria 0001, South Africa; petunia.mengwai@gmail.com (K.M.); perpetua.modjadji@smu.ac.za (P.M.)

**Keywords:** South Africa, non-disclosure, youth, motivation, sexual behaviour, partner reaction, disclosure outcome, implications

## Abstract

The study investigated the motivation to disclose or the decision to withhold one’s HIV serostatus to one’s partners and assessed the implications of non-disclosure on young peoples’ sexual behaviour and access to treatment. This was a cross-sectional survey conducted with 253 youth aged 18–25 years receiving antiretroviral therapy in a health district in North West Province, South Africa. The majority were female (75%), the mean time since the HIV diagnosis was 22 months, 40% did not know their partner’s HIV status, 32% had more than two sexual partners, and 63% had not used a condom during the last sexual act. The prevalence of disclosure was 40%, 36% delayed disclosure for over a year, and most disclosed to protect the partner from HIV transmission, to receive support, and to be honest and truthful. The prevalence of non-disclosure was high, as 60% withheld disclosure due to fear of abandonment, stigma and discrimination, accusations of unfaithfulness, and partner violence. Over half (55%) had no intentions to disclose at all. The lower disclosure rates imply that HIV transmission continues to persist among sexual partners in these settings. The findings suggest that high levels of perceived stigma impact on disclosure and HIV treatment, which increases the risk of on-going HIV transmission among youth receiving long-term antiretroviral therapy (ART) in South Africa.

## 1. Introduction

The HIV epidemic continues to be a major challenge in developing countries, with young people bearing the highest HIV burden. In 2017 the Joint United Nations Programme on HIV and AIDS (UNAIDS) estimated that young people account for a third of new HIV infections globally, and in sub-Saharan Africa (SSA) young women between the ages 15 to 24 account for a quarter of all new HIV infections in the region [1]. Many factors put young people at an elevated risk of HIV infection. While there are biological, social, and economic factors that render young people vulnerable to HIV, early sexual activity and unplanned coerced sexual relationships increase the vulnerability of the youth to HIV infection [2]. Excluding the vertical transmission of mother-to-child transmission of HIV, unprotected sex is the most common route to HIV infection for young people [3].

Individuals diagnosed with HIV have substantial difficulties in telling others that they are infected [4]. This is particularly true with disclosure to sexual partners. Evidence suggests that it is common for people with HIV to not disclose their HIV status to sexual partners [5]. HIV serostatus disclosure is an important component of HIV prevention, with potential benefits for the individual diagnosed with HIV infection and the sexual partner [6]. Furthermore, disclosure is a public health strategy, which motivates sexual partners for early testing, change in sexual behaviour, and early enrolment to antiretroviral therapy (ART) [7,8].

Disclosure empowers a sexual partner to make informed decisions for safer sex practices to reduce new infections [9,10,11], whereas non-disclosure is associated with non-adherence to ART and increases the risk of HIV transmission between sexual partners [9,12]. Non-disclosure is also associated with negative psychosocial effects, such as high stress levels and poor coping skills, due to a lack of partner support [5].

Although the prevalence of HIV status disclosure among adults living with HIV is generally high in South Africa and in SSA [8,13,14,15], young people face challenges over disclosing their HIV status to sexual partners due to their fear of potentially adverse outcomes, such as rejection and stigma [16]. The low disclosure rates among young people living with HIV (YPLHIV) have been associated with the nature of their sexual relationships [4]. Researchers argue that young people’s romantic relationships tend to be more fleeting and sporadic and lack commitment and trust. This leads to the frequent change of sexual partners and engagement in casual sexual contacts or recurrent casual contacts [4,17]. This behaviour makes self-disclosure more complex, as the disclosure of one’s HIV status to a partner might not be a priority when the relationship is fleeting.

While HIV serostatus disclosure is recognised as a cornerstone of HIV management, there is evidence that access and taking HIV treatment alone do not change the environmental factors contributing to the nondisclosure of one’s HIV status [18]. Other contextual factors that determine whether the outcome of self-disclosure is positive or negative are crucial in the decision to disclose [5]. While much literature exists on disclosure among adults, not much has been done about self-disclosure among youth globally [17,19] and within the South African context [20]. Given the potential public health benefits associated with HIV disclosure, it is essential to understand the reasons why young people choose not to disclose their serostatus [16].

The study investigated young people’s decision and motivation to disclose or withhold their HIV serostatus to their sexual partners and assessed the implications of non-disclosure for their sexual behaviour and access to treatment. The results will inform the development of interventions that may facilitate supporting young people through the disclosure process. The study will further highlight the complex and dynamic nature of the disclosure process to increase the awareness of health care professionals who provide services to the growing population of young people living with HIV.

## 2. Methods

### 2.1. Study Design and Setting

The study employed a cross-sectional quantitative descriptive survey that was conducted in community health care (CHC) facilities in Bojanala District in North West Province, South Africa. The district is located 52 km from the city of Pretoria and is classified as a rural district. Nine CHCs were selected from 15 health facilities in the district. All facilities in the district provide HIV testing and counselling and dispense ART to adults and children.

The participants in the study were males and females living with HIV aged between 18 and 25 years who had been attending and receiving ART in the selected CHCs, who had acquired HIV heterosexually, who were currently in sexual relationships, or who had been in a sexual relationship in the past 12 months. The health facilities keep the records of children and adolescents (up to 19 years) with perinatal HIV separate from the general adult population, this assisted the researchers to establish the mode of HIV transmission. In addition, the initial screening for potential participants was done by the facility data capturers and the nurse during consultation with the patients.

The Raosoft sample calculator was used to determine the sample size, using a population size of 1500 at a confidence interval of 95% and a margin of error of 5% with a 50% response distribution rate. A minimum sample of 306 was calculated. However, the total sample for the study was 254 since most patients started to collect their ART from Drug ATMs (a self-service machine where patients obtain their medication in the same way people withdraw money at an ATM) because of the launch of the Central Chronic Medicine Dispensing and Distribution programme (CCMD). The CCMD is a National Department of Health (NDoH) strategy in which stable patients collect their ART from the Drug ATMs. This has reduced the number of patients accessing health facilities for routine ART refill, especially those who are young and employed. Patients who are considered stable get up to six months repeat prescriptions for ART and collect the medication from the Drug ATMs. These patients are required to visit the health facilities for monitoring and renewal of the ART prescriptions after six months. The CCMD was launched in 2018 and significantly reduced the estimated population size from which the sample size for the study was supposed to be drawn. The estimated population of young adults who were on the data base before the launch was 1500, and, as stated, the calculated sample size could not be attained. Nevertheless, the researcher stayed in the field long over three months to conduct a census of all the patients who remained in the data base after the launch. We used an appointment system and convenient sampling to recruit participants who met the inclusion criteria when they came for repeat prescriptions, monitor viral load, those who had not met the criteria to use the Drug ATM, and those who came for their routine ART refill and consultation.

### 2.2. Data Collection

A structured and pre-tested, self-administered, and/or researcher-administered questionnaire was used to collect data. The questionnaire had been adapted from a previous similar study [8]. It was first prepared in English and then translated into the local Setswana language. The information collected included socio-demographics, partnership status, sexual behaviour, HIV clinical variables, and disclosure of HIV status of the respondents.

The outcome variable was disclosure to current sexual partner, which included steady and casual relationships, as well as other sexual partners in the past 12 months for those in multiple sexual partnerships. The secondary outcome variables were: the time it took to disclose, the reasons for disclosure, the outcome of disclosure, the reasons for non-disclosure, and the intentions to disclose in the future. Partnership status was a key explanatory variable, we measured the nature and duration of the relationship, multiple partnerships, and the living arrangements of the couple. We further established whether this was the partner they had when they tested positive, the number of times they changed a partner after testing positive, and concurrent sexual relationships. Another key explanatory variable was the use of a condom during last sexual encounter.

A trained research assistant and the lead author collected the data. Training for the research assistant on the purpose of the study, the questionnaire, voluntary participation, and ensuring privacy and confidentiality was done. All data were collected in the selected facilities, in a private room for confidentiality, mostly in the mornings or afternoons, depending on the availability of the participants at the time.

### 2.3. Data Analysis

After the data had been collected, the raw data were captured in an Excel spreadsheet, cleaned, and coded. The data were then imported to Stata version 13 for analysis. Descriptive statistics were performed to summarise all the categorical variables. The Pearson Chi^2^ test was used to examine the associations between HIV disclosure and the explanatory variables, such as age, gender, duration of partnership, duration of ART, educational status using bivariate analysis. The means of continuous variables were computed using the student *T*-Test. Significance was set at *p* < 0.05.

### 2.4. Ethics

Ethical approval was obtained from the Research Ethics Committee of Sefako Makgatho Health Sciences University [SMUREC/H/88/2018: PG]. The study received the permission from the relevant authorities in Bojanala Health district. Written consent was obtained from all the participants in the study, and they were assured of their right to withdraw from the interview if they wished, and of their anonymity.

## 3. Results

### 3.1. Sociodemographic Characteristics

The socio-demographics of the participants are summarised in Table 1. A total sample of 253 YPLHIV was obtained from the selected CHCs. There were 190 (75%) females and 63 (25%) males, with a mean age of 22 years (SD = ±2). The age was categorised into two groups: 18–21 years and 22–25 years; 51% were aged between 22 and 25 years and 49% were between 18 and 21 years. The participants were characterised by unemployment (72%), having no income (74%), being single (86%), and having completed the 12th grade (45%). The majority (76%) were not living with their current sexual partners, 65% reported that their current partners were boyfriends, 29% were girlfriends, 2% were spouses, and 4% were currently not in a sexual relationship.

The difference between participants who disclosed and those who did not were significant for gender (*p* ≤ 0.0001), age (*p* ≤ 0.0001), marital status (*p* = 0.002), employment (*p* = 0.008), education (*p* = 0.000), and living arrangements (*p* = 0.049).

### 3.2. Sexual Behaviour and Relationship Characteristics

Table 2 shows the sexual behaviour of the study participants. Ten (4%) of the participants were not in a sexual relationship at the time of data collection, 37% were in relatively new sexual relationships of less than a year, 39% had been in a relationship with the current partner for up to three years, and 24% for more than three years. A large proportion (42%) had changed their partners after testing HIV positive, while 58% had remained in the same sexual relationship. Concerning their sexual practice in the previous 12 months, 68% had had one sexual partner, 32% had had two or more sexual partners, and 34% had had more than two sexual partners at the same time. With regard to condom use, 63% had not used a condom during their last sexual encounter.

Differences in sexual behaviour between those who disclosed and those who had not disclosed were observed for refusing sex when a condom was not used (*p* ≤ 0.0001).

### 3.3. HIV Clinical Data Characteristics of YPLHIV

The related clinical data are presented in Table 3. The duration of knowing the HIV diagnosis was categorised as ≤ 1 year, 2–4 years, and ≥ 5 years. The majority (52%) had known about their HIV diagnosis for 2–4 years. The duration on ART was divided into two categories: ≤ 2 years, > 2 years. The majority (75%) had been receiving ART for ≤ 2 years. Concerning the partner’s status, 40% did not know their partner’s HIV status. Of those who knew their partner’s status, 37% reported that their partners were HIV negative. The differences between participants who disclosed and those who did not disclose were significant for time since diagnosis (*p* ≤ 0.0001), duration on ART (*p* ≤ 0.0001), and knowledge of partners’ status (*p* ≤ 0.0001).

### 3.4. Disclosure Status of YPLHIV and Motivation

Table 3 describes the disclosure characteristics of the participants. Overall, 60% reported that they had not disclosed their status to their partners. Additionally, disclosure to multiple sexual partners was low; slightly over a quarter (28%) of those who had disclosed reported that they also disclosed to the other partners. Of the 40% who had disclosed to their partners, 46% had disclosed immediately, whereas 36% had taken more than a year to disclose. The proportion of those who disclosed was higher among those with a longer time since HIV diagnosis (2–4 years) than among those with a shorter duration of HIV diagnosis (≤1 year) (47 (50%) vs. 29 (31%)), and higher among those with shorter duration of ART (≤2 years) than among those with a longer duration (>2 years) (57 (61%) vs. 37 (39%)).

Table 4 describe the disclosure status and motivation to disclose. The main motivation given for disclosing was to protect their partners from HIV transmission (40%), to be truthful and honest to their partners (21%), and to get support from their partners (17%), whereas the main reasons for not disclosing their status were fears that their partners might leave them (34%), fear of stigma and discrimination (27%), fear that their partners might accuse them of being unfaithful (16%), and fear that their partners would be angry with them (14%).

With regard to the intention to disclose, 55% of those who had not disclosed had no intentions to disclose, and 80% of those with intentions to disclose did not know when disclosure will occur. With regard to the partners’ reactions to the disclosure event, the results showed that 44% had denied the test results, 33% were shocked, 11% were angry, and 8% were supportive. Concerning explaining the taking of ART to partners in the absence of disclosure, 37% were hiding the ART, 25% were lying about taking ART, and 38% gave other reasons.

## 4. Discussion

We investigated the decision and motivation for disclosure and the reasons for not disclosing their HIV status to their sexual partners among YPLHIV and assessed the public health implications of non-disclosure. The majority (75%) of the participants aged 18–25 years had been receiving ART for less than two years. They accessed ART in rural health facilities in Bojanala Health district. The study found low disclosure rates to partners. Only 40% of the participants reported to have disclosed their HIV status to their partners. This indicated a high prevalence of non-disclosure, as 60% had delayed disclosure. Disclosure had been delayed despite significant proportion (58%) of YPLHIV remaining in the relationship with the sexual partner after testing positive. Similar rates of delayed disclosure to sexual partners were found in other studies [21,22,23,24]. Lower disclosure rates imply that HIV transmission will continue to persist among sexual partners in these and similar settings [25]. In 2018, South Africa had an estimated 7.7 million people living with HIV and young adults aged between 15–24 years accounted for 18.8% of new infections [1].

The disclosure prevalence reported in the current study is lower than the rates of 44% to 68% observed in other studies in SSA [9,15,26,27]. Earlier studies also reported high prevalence ranging from 75%–97% among adults [14,28,29,30]. The inconsistency observed could be partly explained by the difference in the study population: while all the other studies were conducted with adults on long-term ART, the current study was conducted with youth with a mean age of 22 years.

Non-disclosure was informed by several reasons including fear of rejection, fear of stigma and discrimination, fear of abandonment, and fear of being accused of being unfaithful. These findings are consistent with those of other previous studies [8,11,31]. While knowing HIV status and/or receiving ART for a long time is associated with disclosure to sexual partners in adult PLHIV [8,14], this was not the case in this study. Disclosure was withheld even among those that had known about their HIV-positive status for over two years.

The decision to disclose one’s seropositive status is a complex process influenced by multiple socio-cultural beliefs, factors, and personal circumstances of the participants [32]. In the current study, the prevalence of disclosure was 40%, and it was motivated by the need to protect the partners from HIV transmission, to receive support from their partners, and to be honest and truthful to their partners. Consistent with other studies, concern about protecting the sexual partner from HIV infection was the most common motivator for disclosing [8,11,33]. Despite the disclosures being motivated by the need for support, less than a tenth of the participants in the study had received support from the partner following disclosure.

The study further found that a significant proportion (40%) of the participants did not know their partner’s HIV status. This explains the low rate of disclosure observed in the study. Evidence suggest that knowing the HIV status of a sexual partner increases the chances of disclosure between sexual partners [13,34]. Disclosure between partners benefits the HIV-negative partner because it may result in the adoption of safer sexual practices and behaviour in the relationship to prevent HIV transmission [35]. This is crucial in cases such as those in the current study because, as indicated, a significant proportion of the young people did not know their partners’ HIV status.

Disclosure for over a third (36%) of those who had disclosed was delayed to over a year after the HIV-positive test results. Researchers suggest that young people are unlikely to disclose their HIV status within a short-term romantic relationship [17,36]. However, the disclosure was low despite the duration of their ART. For more than half of participants it was over one year. The fear of abandonment by the sexual partner, the fear of stigma and discrimination, the fear of an accusation of unfaithfulness, as well as the fear of violence were the main deterrents against disclosure in the current study. Similar findings have been reported elsewhere [11,26,31,37]. The implications of the delay in disclosure are that it is unlikely that safe sex practices will be adhered to, with possible consequences for HIV transmission to the HIV negative partner, and increased reinfection among HIV-positive partners [21,22]. The population of young people is particularly vulnerable because of the nature of their short-lived romantic relations, which do not prioritise disclosure to the romantic partner. In the current study, a significant proportion (42%) of the participants had changed their partners after testing HIV positive.

It is worrying that over two thirds (63%) of the participants reported practising unprotected sex with their partners in their last sexual encounter despite the majority being in steady relationships of more than one year. The literature shows that in SSA approximately two thirds of HIV incidence occurs in steady partnerships [38], a fact which underscores the importance of disclosure among sexual partners. Delayed disclosure and unprotected sex occurred within a context where 40% of the participants did not know their partner’s HIV status. Knowing the sexual partner’s HIV status is a significant predictor of safer sexual practices, such as condom use and a reduction of the number of sexual partners [12]. This finding highlights the public health implications of the increased risk of on-going HIV transmission among YPLHIV receiving long-term ART in SSA. Therefore, the need to promote and strengthen HIV testing among partners cannot be over-emphasised.

The high prevalence of multiple sexual partnerships coupled with the considerable prevalence of unprotected sex among this population of young people have significant implications for HIV prevention interventions. It is more worrying that multiple sexual partnerships were common among disclosed partners. Consistent with previous findings by Mkhwanazi et al. [39], disclosure did not lead to low-risk behaviours in the current study. The researchers highlighted the low incidence of using condoms during sexual intercourse among partners who did not disclose [10,40]. However, the observation is contrary to those recorded in previous findings, where the researchers observed a reduction in risk behaviour following disclosure to sexual partners [9,10,41].

Concerning partner reactions to disclosure, a significant proportion of the participants experienced negative partner reactions. They reported that partners denied the positive test results (45%) and reacted with shock (34%) and anger (10%) after the disclosure event. This observation was consistent with those in other studies from SSA [11,21,22,37]. However, none of the participants reported physical violence and abandonment after disclosure [22] and only a small proportion (7%) of the partners reacted positive to disclosure and were supportive towards their partners.

The intention to disclose the HIV serostatus to the partner in the future was relatively rare and over half (55%) of the participants had no intentions to disclose at all. The disclosure of one’s HIV-positive status is a sensitive issue for PLHIV, who need to weigh the benefits against the harm before disclosing their status [24]. This suggests that the fear of the perceived negative consequences of disclosure outweighed the benefits of disclosure for the participants in the current study. This is evident in the fact that the few who intended to disclose had not thought out a plan or a time to disclose. Other studies have suggested that the nature of the sexual relationship one has influences the intention to disclose to partners [4,42]. Nevertheless, researchers suggest that the likelihood of disclosure is likely to increase with the duration of living with HIV [41,43].

Non-disclosure resulted in lying about clinic attendance and ART refills, withholding reasons for the check-ups, and hiding the ART medication from their partners. Other researchers reported similar consequences of concealing one’s HIV status from a partner, such as avoiding care, and/or not adhering to ART [10]. The findings highlight the lack of communication about HIV and ART, which increases the need to take medication in private. According to Trinh et al. [44], disclosure can provide an open environment where pill taking need not be concealed. Hiding medications and other actions aimed at concealing one’s HIV serostatus from sexual partners and/or family members leads to poor ART adherence [45]. Evidence suggests that many PLHIV in South Africa experience high levels of perceived and enacted stigma, which impede HIV treatment and HIV disclosure [46].

### Study Limitations

A major limitation of the study is the small sample size that limits the ability to generalize to young people in other settings. In addition, the study setting was rural and may not be representative of young people in urban settings. The study findings are subject to social desirability, the use of researcher administered tool might have influenced the participants to over or under report their responses. To mitigate this bias, the researcher’s assurance of the participants’ confidentiality and anonymity was maintained whereby the participant’s personal identifier was not captured. Lastly, we did not collect other clinical variables, particularly the viral load to assess the relation with disclosure.

## 5. Conclusions

The study found a low rate of HIV disclosure to sexual partners, with over half the participants delaying disclosure to over a year after the HIV-positive test results. The delay in disclosing was not aligned with the main motivation to disclose that was cited, because while most wanted to protect the partners from HIV transmission and to be honest and truthful to their partners, disclosure was delayed.

The same was true of the intentions to disclose, as more than half of the participants had no intentions to disclose at all. This implies that the deterrents against disclosure are real concerns for YPLHIV in these settings.

In contrast, the prevalence of non-disclosure was high and was informed by the fear of rejection, the fear of stigma and discrimination, the fear of abandonment, and the fear of being accused of being unfaithful. Non-disclosure resulted in thier lying about clinic attendance and ART refills, withholding the reasons for their check-ups, hiding their ART medication from their partners, and engaging in multiple sexual partnerships and unprotected sex. Non-disclosure and high levels of perceived stigma impact on HIV treatment and prevention and result in an increased risk of HIV transmission among sexual partners.

There is a need for the health care professionals who provide services to the increasing population of YPLHIV to be involved in the disclosure process, as they are in a better position to understand the complex nature of the disclosure process for this population group.

Given the complex dynamics that contribute to a high rate of risky sexual behaviour among PLHIV, we recommend that the National Department of Health facilitate the development of effective interventions for safe sexual practices early in HIV care and disclosure to contribute to HIV prevention, since disclosure correlates with greater condom use.

## Figures and Tables

**Table 1 healthcare-08-00253-t001:** Socio-demographic characteristics of youth living with HIV by disclosure status.

Variables	All *n*(%)	Disclosed *n*(%)	Non-Disclosed *n*(%)	*p*-Value
**Gender** Female Male No partner (*n* = 10)	190 (75) 63 (25)	87 (90) 10 (10)	97 (66) 49 (34)	≤0.0001
**Age** 18–21 years 22–25 years No partner (*n* = 10)	120 (49) 123 (51)	82 (85) 15 (15)	38 (26) 108 (74)	≤0.0001
**Marital status** Single Cohabiting Married **No partner (*n* = 10)**	209 (86) 31 (12.8) 3 (1.2)	90 (93) 4 (4) 3 (3)	119 (82) 27 (18) 0 (0)	0.002
**Employment status** No Yes **No partner (*n* = 10)**	175 (72) 68 (28)	79 (81) 18 (19)	96 (66) 50 (34)	0.008
**Education** No school/primary school Did not complete 12th grade Completed 12th grade Tertiary education No partner (*n* = 10)	24 (10) 91 (38) 110 (45) 18 (7)	5 (5) 53 (55) 29 (30) 10 (10)	19 (13) 38 (26) 81 (55) 8 (6)	0.000
**Income** No income <R2000 R2000–R5000 >R5000 No partner (*n* = 10)	187 (74) 27 (11) 33 (13) 6 (2)	76 (82) 7 (8) 8 (9) 1 (1)	98 (67) 19 (14) 24 (16) 5 (3)	0.069
**Living arrangement** Not living with a sexual partner Living with a sexual partner No partner (*n* = 10)	184 (76) 59 (24)	68 (70) 29 (30)	116 (79) 30 (21)	0.049

**Table 2 healthcare-08-00253-t002:** Sexual behaviour and relationship characteristics of YPLHIV by disclosure status (*n* = 143).

Variables	All *n*(%)	Disclosed *n*(%)	Non-Disclosed *n* (%)	*p*-Value
**Changed partners after testing HIV positive** No Yes	141 (58) 102 (42)	62 (64) 35 (36)	79 (54) 67 (46)	0.129
**Sexual partner(s) in 12 months (*n* = 233)** One Two More than two	164 (68) 57 (23) 22 (9)	71(73) 21 (22) 5 (15)	93 (64) 36 (25) 17 (11)	0.234
**Concurrent sexual relationship** No Yes	166 (66) 77 (34)	68 (72) 26 (28)	93 (66) 47(34)	0.603
**Duration of sexual relationship (missing *n* = 10)** <1 year Between 1 and 3 years >3 years	83 (37) 86 (39) 54 (24)	40 (45) 28 (32) 20 (23)	43 (32) 58 (43) 34 (25)	≤0.106
**Condom use in last sexual act (missing *n* = 8)** No Yes	148 (63) 87 (37)	56 (61) 36 (39)	92 (64) 51 (36)	0.680

The statistic excludes 10 participants who were not in a sexual relationship during data collection.

**Table 3 healthcare-08-00253-t003:** HIV clinical data of YPLHIV (*n* = 243).

Variables	*n*	Disclosed *n*(%)	Non-Disclosed *n*(%)	*p* Value
**Time since HIV diagnosis (missing *n* = 3)** ≤1 year 2–4 years ≥5 years	90 (38) 125 (52) 25 (10)	29 (31) 47 (50) 18 (19)	61 (42) 78 (53) 7 (5)	≤0.0001
**Duration on ART (missing *n* = 5)** ≤2 years >2 years	179 (75) 59 (24)	57 (61) 37 (39)	122 (84.7) 22 (15.3)	≤0.0001
**Know partner’s HIV status (*n* = 158)** No Yes	99 (40) 144 (60)	3 (4) 94 (96)	96 (66) 50 (34)	≤0.0001

**Table 4 healthcare-08-00253-t004:** Disclosure status and motivation to disclose to sexual partners.

Variables	*n*	%
**Disclosure of HIV status** No Yes No partner (*n* = 10)	146 97	60 40
**Time taken to disclose (*n* = 97)** Immediately >1 year Tested together with partner Other times	45 35 10 7	46 36 11 7
**Motivation to disclose (*n* = 97)** Protect partner from HIV transmission To be honest and truthful to the partner To get partner support Tested with their partners Other reasons	39 20 16 10 12	40 21 17 10 12
**Partner’s reaction to disclosure (*n* = 97)** Denied test results Shocked Angry Supportive Other reasons	42 33 11 7 4	45 34 10 7 4
**Reasons for non-disclosure (*n* = 146)** Fear of abandonment Accusation of unfaithfulness Partner will be angry Fear of stigma and discrimination Other	49 23 20 39 15	34 16 14 27 10
**Intentions to disclose in the future** No Yes	82 65	55 45
**Plans to disclose in the future** Not sure when to disclose Within 12 months Within six months	57 8 6	80 11 9

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
