# Peer review of "Low Disclosure Rates to Sexual Partners and Unsafe Sexual Practices of Youth Recently Diagnosed with HIV; Implications for HIV Prevention Interventions in South Africa"

_healthcare, 2020, doi:10.3390/healthcare8030253_

Round 1

Reviewer 1 Report

This was a cross-sectional survey conducted with 253 youth aged 18-25 years receiving ART in a health district in North West Province, South Africa. Overall, I found the paper well-organized and interesting.

Major comments

  1. How disclosure was measured in the present study, especially when participants had multiple sex partners? Did you measure disclosure to the most recent partner? Or the partner in a stable relationship?
  2. As all participants were on ART, how about their viral load? As undetectable viral load equals no HIV transmission, what is the point of discussing disclosure/non-disclosure in the context of undetectable viral load?
  3. The authors should provide more details on the measurement used in the present study in the “Data collection” section.
  4. Limitations of the present study should be discussed.
  5. In addition to disclosure/non-disclosure, intention to disclose might be another interesting outcome to explore.

Minor comments

  1. Line 83 How acquired HIV heterosexually was confirmed in the present study?
  2. I did not fully understand the power estimation. For the number 306, was disclosure rate in previous studies being referred to? Was the final sample sufficient or not?
  3. Line 98, if the questionnaire was researcher-administered, how about self-reported bias?
  4. In Table 2, it is a bit hard to interpret “duration of sexual relationship”. Was that an exploration about a longer duration of relationship would facilitate disclosure? As this was a cross-sectional study, any casual inference should be very careful.

Author Response

We thank the reviewer for the valuable comments on our manuscript. We are have addressed all the comments to the best of our ability in the relevant sections of the manuscript. Below is a line by line response to the comments.  

How disclosure was measured in the present study, especially when participants had multiple sex partners? Did you measure disclosure to the most recent partner? Or the partner in a stable relationship?

Response: We measured disclosure to the current partner as well as to the other sexual partners in the past 12 months. Besides defining the nature of the relationship, we also measure the duration of the current relationship, the living arrangements of the couple, established whether this was the partner they had when tested positive, and the number of time they changed a partner after testing positive.

Concerning disclosure in multiple sexual partnership, 27% of those who disclosed also disclosed to their other sexual partners, these data include those who had more than two partners in the past 12 months (table 2). We added text in the methods section to describe the disclosure related variables that were measured. We explain that the outcome variable was disclosure to current partner, which included steady and casual relationships, as well as other sexual partners in the past 12 months for those in multiple sexual partnership. The secondary outcome variables were; the time it took to disclose, the reasons for disclosure, the outcome of disclosure, the reasons for non-disclosure, and the intentions for disclosure in the future.

As all participants were on ART, how about their viral load? As undetectable viral load equals no HIV transmission, what is the point of discussing disclosure/non-disclosure in the context of undetectable viral load?

Response: In South Africa, the viral load is assessed at 6 months, 12 months, and 18 months for patients initiated through the universal treat all strategy. We did not collect information on the viral load as an explanatory variable, our experience from using patients records in primary health facilities has shown that record keeping is very poor particularly the filling of test results, furthermore, in the treat all strategy, people are initiated with high CD4 count and viral load. We acknowledge the scientific evidence that undetectable viral load equals no HIV transmission, however, the study not taken that into consideration during conceptualization. This has been reported as a limitation of the study. We were motivated to carry out this study because disclosure is a public health strategy which motivates sexual partners for early testing, change in sexual behaviour, and early enrolment to ART.

The authors should provide more details on the measurement used in the present study in the “Data collection” section.-see relevant section under data collection line 113-121 page 3

Limitations of the present study should be discussed

Response: We added study limitations

In addition to disclosure/non-disclosure, intention to disclose might be another interesting outcome to explore.

Response: We measured intentions to disclose and 55% of those who had not disclosed had no intentions to disclose in the future, and 80% of those with intentions to disclose had not made up their mind when to disclose. There were no differences in gender, nature of relationship, duration of relationship, duration on ART, age, and intention to disclose table 4.

Minor comments

Line 83; how acquired HIV heterosexually was confirmed in the present study?

Response: We added text from line 85-88 page 2 explaining the process to confirm the mode of HIV infection of the study participants.

I did not fully understand the power estimation. For the number 306, was disclosure rate in previous studies being referred to? Was the final sample sufficient or not?

Response: The sample size was estimated using a sample size calculator (The Raosoft sample calculator) We explain the challenges of meeting the estimated sample size after the launch of Drug ATMs by the South Africa National Department of Health as a programme in which stable patients collect their ART from the Drug ATMs and not the clinics anymore, this reduced the pool from which to sample and conducted a census of all those who remained in the data base of the facilities. We explain this from line 89-106 page 2-3.

Line 98, if the questionnaire was researcher-administered, how about self-reported bias?

Response: Social desirability has been reported as a potential bias in the study limitations

In Table 2, it is a bit hard to interpret “duration of sexual relationship”. Was that an exploration about a longer duration of relationship would facilitate disclosure? As this was a cross-sectional study, any casual inference should be very careful.

Response: We asked the participants to indicate how long they have been in the relationship. Research shows an association between partnership duration and disclosure.

Reviewer 2 Report

In my mind the manuscript is absolutely interesting and worth reading. The results are exciting and shocking. It is about whether and when young people tell their partner that they are diagnosed as HIV positive and the article helps to understand the motives, why many infected do not tell their partner the truth. 61% of the participants did not tell their partner that they are HIV positive! It took 34% of the other group more than a year to tell the truth. Despite HIV infection, 63% did not use a condom during their last sexual intercourse and 66% would not refuse to have sex, even if no condom was on hand.

This information can be useful to understand the motives and to get young people to tell their partner the truth more often and to prevent their partner from becoming infected.

Unfortunately, the article suffers from many small mistakes:

Using a case number calculation, the authors came to a minimum of 306 participants; unfortunately the study only included 253 participants. In my eyes, this is still quite a large sample.

For "young people living with HIV" is partly the abbreviation YPLHIV used, partly also YLHIV.

Line 123 : Here it is said that 186 (74%) were female and 67 (26%) were male = 253. In Table 1 there are 135 (71%) female and 54 (29%) male. That is only 189.

Line 124: How were the age groups chosen? The authors separate 18-20 (38%) and 21-25 years (62%). I have no logical reason why the age limit was chosen in this way? Please give an explanation.

Line 126:  87% are single. I don't understand if you don't have a partner, how can these participants have a coming-out to their partner? Although 87% are single, 76% do not live with their partner. How does it work?

Line 127: What exactly does "secondary education" mean? Givbe a number of school years for readers not familiar with this system of education.

Lines 127-129: 54.8% have boyfriends as partners, 31.4% have a girlfriend. That is 86.2%. And the others?

Tab. 1: Which statistical test was used to calculate the p-values?

The dates are incorrect. Marital status: 219 + 31 = 250 (it should be 253); thereof 89 disclosed and 146 non-disclosed = 235.

Employment Status: Total 253 but 95 disclosed + 146 non-disclosed = 241.

Education: Total 253 but disclosed 92 + non-disclosed 146 = 238.

Etc.

Most of the numbers in the disclosed and non-disclosed columns do not match the total in column 2.

Tab. 2: Here are the same errors. For example:

Changed partners after testing HIV positive: A total of 244 (it should be 253), of which 96 + non-disclosed 149 = 245.

Etc. in the other lines.

Line 151: Two categories are distinguished here: greater than / equal to 2 years and greater than two years. What is the difference?

In contrast to the text, Table 3 then differentiates three age groups for the length of time since diagnosis: less than or equal to 1 year, 2-4 years, over 5 years.

Line 160: The numbers 90% women and 10% men in the group of disclosure participants cannot be compared in this way, since there are far fewer men in the study. 9 out of 54 men are 16% and not 10 who told their partner. 82 out of 135 women are 60% and not 90%. Since the distribution of men and women is very uneven, the raw values ​​cannot be taken as part of the overall group.

Table 4: For Non-Disclosure of HIV status, the number of 146 participants is given; in Table 1, however, there were only 98 participants, in the disclosure there are 92 participants, in Table 1, however, 91. The same applies to the other values. There are completely different numbers of participants between 72 and 146.

It would be nice to have figures/graphs of the data in addition to tables.

In the discussion it would be interesting to know how many HIV-infected people are there in South Africa? How is the development? To what extent are the numbers increasing? How many people die from HIV?

Author Response

We appreciate and thank the reviewer for a thorough review of our manuscript. We revised the results and provided detailed response to all the comments in the table below.

Reviewer comments

Response

Using a case number calculation, the authors came to a minimum of 306 participants; unfortunately the study only included 253 participants. In my eyes, this is still quite a large sample.

For "young people living with HIV" is partly the abbreviation YPLHIV used, partly also YLHIV.

We used YPLHIV throughout the document, the most common abbreviation approved by the UNAIDs is for adolescents living with HIV (ALHIV), our sample included young people

Line 123 : Here it is said that 186 (74%) were female and 67 (26%) were male = 253. In Table 1 there are 135 (71%) female and 54 (29%) male. That is only 189.

We cleaned the data again and reanalysed to identify the discrepancies and made the necessary corrections in the tables and descriptions throughout the document including the discussion. All the changes are in track changes. .

Line 124: How were the age groups chosen? The authors separate 18-20 (38%) and 21-25 years (62%). I have no logical reason why the age limit was chosen in this way? Please give an explanation.

Thanks for the comment, we used the mean and median as a cut off for the age categories (18-21 and 22-25) and reanalysed the data.

Line 126:  87% are single. I don't understand if you don't have a partner, how can these participants have a coming-out to their partner? Although 87% are single, 76% do not live with their partner. How does it work?

The participants were asked to indicate their marital status and 87% were single, 12% were cohabiting (living with the partner out of marriage), and 1% was married. The selection criteria did not exclude participants who were not in sexual relationship at the time of data collection. The argument was that their relationship status could have changed after testing positive and they would still inform the study on issues related to reasons for disclosure and non-disclosure, outcome of disclosure, and intention to disclose. The ten participants who were currently not in sexual relationship are highlighted in the tables.

Line 124: How were the age groups chosen? The authors separate 18-20 (38%) and 21-25 years (62%). I have no logical reason why the age limit was chosen in this way? Please give an explanation.

Thanks for the comment, we used the mean and median as a cut off for the age categories and reanalysed the data.

Line 127: What exactly does "secondary education" mean? Give a number of school years for readers not familiar with this system of education

Response: The educational status of the participants show four categories, primary schooling, completed the 12th Grade, tertiary education and a group that completed primary school but did not completed the 12th grade.

Lines 127-129: 54.8% have boyfriends as partners, 31.4% have a girlfriend. That is 86.2%. And the others?

We reanalysed the data and the results now show that those that had a spouse and those who were not in a sexual relationship

Tab. 1: Which statistical test was used to calculate the p-values?

The Pearson Chi2 test was used to examine the associations between HIV disclosure and the explanatory variables-this is added under data analysis-line 130-133 page 3

The dates are incorrect. Marital status: 219 + 31 = 250 (it should be 253); thereof 89 disclosed and 146 non-disclosed = 235.

Corrected: The marital status results (table 1) show that;  Single (209), Cohabiting (31), Married (3), and no partner (10), this add up to 253

Employment Status: Total 253 but 95 disclosed + 146 non-disclosed = 241.

Corrected; in table 1, the results also show those without a sexual partner (n=10).

Education: Total 253 but disclosed 92 + non-disclosed 146 = 238

Corrected; in table 1, the results also show those without a sexual partner

Most of the numbers in the disclosed and non-disclosed columns do not match the total in column 2.

We have reanalysed the data and added those who were currently not in a sexual relationship. All the disclosed and non-disclosed column excludes this group.  

Line 151: Two categories are distinguished here: greater than / equal to 2 years and greater than two years. What is the difference?

This is a typo, it is supposed to be less/equal to 2 years and greater than 2 years

Tab. 2: Here are the same errors. For example:

Changed partners after testing HIV positive: A total of 244 (it should be 253), of which 96 + non-disclosed 149 = 245.

All the results in the tables now show the 10 participants who were currently not in a sexual relationship.

In contrast to the text, Table 3 then differentiates three age groups for the length of time since diagnosis: less than or equal to 1 year, 2-4 years, over 5 years.

The time since HIV diagnosis and the duration on ART may differ for people who not were initiated using the universal test and treat strategy, this informed the decision to measure both.

Line 160: The numbers 90% women and 10% men in the group of disclosure participants cannot be compared in this way, since there are far fewer men in the study. 9 out of 54 men are 16% and not 10 who told their partner. 82 out of 135 women are 60% and not 90%. Since the distribution of men and women is very uneven, the raw values cannot be taken as part of the overall group.

We used the column to compare the group for all the variables. In this case we compared males and females. Your suggestion is that we use row comparisons instead

Table 4: For Non-Disclosure of HIV status, the number of 146 participants is given; in Table 1, however, there were only 98 participants, in the disclosure there are 92 participants, in Table 1, however, 91. The same applies to the other values. There are completely different numbers of participants between 72 and 146.

We reanalysed the data and corrected the discrepancies in table 4

In the discussion it would be interesting to know how many HIV-infected people are there in South Africa. How is the development? To what extent are the numbers increasing? How many people die from HIV?

We included the HIV stats in the discussion line 213-215.

Round 2

Reviewer 1 Report

I am satisfied with the response and the revised manuscript. Well done!

Author Response

Thanks for the positive comments about the revised manuscript 

Reviewer 2 Report

The manuscript is now much better and absolutely exciting to read. I have only two small things to criticize: It is called Chi² with superscript "2", not Chi2. Some of the numerical values do not add up to 100, but only to 99% due to rounding errors. For example: 49 out of 146 are 33.56 = 34%. 108 out of 146 are 73.97% = 74%. The authors should check that again.

Otherwise the article is ready for printing in my opinion.

Author Response

The manuscript is now much better and absolutely exciting to read. I have only two small things to criticize: It is called Chi² with superscript "2", not Chi2. Some of the numerical values do not add up to 100, but only to 99% due to rounding errors. For example: 49 out of 146 are 33.56 = 34%. 108 out of 146 are 73.97% = 74%. The authors should check that again.

Response

Thanks for pointing out this errors

We checked all the tables and made corrections on numerical values that did not add up to 100%. These are highlighted in blue in the tables.

We also changed Chi2 to Chi2 as pointed out.